# Combinatorial programming of human neuronal progenitors using magnetically-guided stoichiometric mRNA delivery

Sayyed M Azimi[1], Steven D Sheridan[1,2], Mostafa Ghannad-Rezaie[1,3], Peter M Eimon[1], Mehmet Fatih Yanik[1,3]*

[1]Department of Electrical Engineering and Computer Science, Massachusetts Institute of Technology, Cambridge, United States; [2]Center for Genomic Medicine, Massachusetts General Hospital, Harvard Medical School, Boston, United States; [3]Department of Information Technology and Electrical Engineering, Swiss federal Institute of Technology Zurich (ETH), Zurich, Switzerland

**Abstract** Identification of optimal transcription factor expression patterns to direct cellular differentiation along a desired pathway presents significant challenges. We demonstrate massively combinatorial screening of temporally-varying mRNA transcription factors to direct differentiation of neural progenitor cells using a dynamically-reconfigurable magnetically-guided spotting technology for localizing mRNA, enabling experiments on millimetre size spots. In addition, we present a time-interleaved delivery method that dramatically reduces fluctuations in the delivered transcription factor copy numbers per cell. We screened combinatorial and temporal delivery of a pool of midbrain-specific transcription factors to augment the generation of dopaminergic neurons. We show that the combinatorial delivery of *LMX1A*, *FOXA2* and *PITX3* is highly effective in generating dopaminergic neurons from midbrain progenitors. We show that *LMX1A* significantly increases *TH*-expression levels when delivered to neural progenitor cells either during proliferation or after induction of neural differentiation, while *FOXA2* and *PITX3* increase expression only when delivered prior to induction, demonstrating temporal dependence of factor addition.
DOI: https://doi.org/10.7554/eLife.31922.001

*For correspondence: yanik@ethz.ch

Competing interests: The authors declare that no competing interests exist.

## Introduction

The human nervous system contains at least hundreds of distinct subtypes of neurons that are necessary to create complex circuits. Generation of these specific neuronal cell types from progenitor cell populations in vitro is important for basic science, drug screening, and potentially for cell-based therapeutics.

Significant effort has shown that known extracellular trophic factor cocktails, although useful, do not enable sufficient control and specificity to efficiently reprogram progenitor populations to a desired differentiated state (*Kriks et al., 2011*; *Maroof et al., 2013*; *Shi et al., 2012a*; *Shi et al., 2012b*). It should be possible to systemically screen for transcription factor regimes that steer the fate of differentiating cells by ectopically introducing factors in a combinatorial manner. In fact, the pioneering work of Yamanaka demonstrated the potential of such an approach by reprogramming somatic cells to pluripotency (*Takahashi and Yamanaka, 2006*).

Although neurons inherit the same genes, cell fate decisions during neurogenesis are mediated through the unique and highly coordinated temporal and spatial expression of hundreds of transcription factors. The efficiency of lineage-specific differentiation can likely be greatly increased if key transcription factor combinations are delivered to cells in a stoichiometrically and temporally optimized manner over the course of differentiation. For instance, while the expression of early genes is

necessary for differentiation and expansion of neuronal progenitors, these same factors may lead to malformations if not silenced later. Similarly, terminal differentiation factors might have deleterious effects if introduced too early.

One could envision using plasmid or non-replicative viral vectors to deliver transcription factors transiently, relying on cell division to dilute out undesired factors with time. However, this approach has severe limitations: (1) it affords only crude temporal control over transcription factor levels and stoichiometries while depending on a proliferative cellular state, (2) the copy numbers delivered to the cells are highly variable, and (3) the use of any DNA or integrative viral vector entails the risk of integration-induced mutagenesis leading to functional genetic alterations. Any protocol based on such techniques could not be risk free or cost effectively translatable to clinical practice. Chemically inducible vectors such as doxycycline-regulated systems also face similar challenges.

In vitro synthesized mRNAs represent an important alternative to the use of DNA vectors (*Karikó et al., 2005*; *Karikó and Weissman, 2007*; *Karikó et al., 2008*; *Angel and Yanik, 2010*; *Karikó et al., 2012*; *Kormann et al., 2011*; *Mandal and Rossi, 2013*; *Warren et al., 2010*). By suppressing the innate immune response to reduce the toxicity of exogenous RNA, we have previously been able to achieve high expression levels of transcription factors in cells with repeated delivery while avoiding the risk of genomic insertion (*Angel and Yanik, 2010*). In addition, since the lifetime of mRNA is shorter than that of DNA-based vectors, the temporal availability of transfected RNAs can be much more precisely controlled over repeated delivery.

Although several transcription factor libraries exist, and significant amounts of microarray and in situ hybridization data are now available, even for the relatively well-studied neuron types, testing all possible stoichiometries and temporal delivery patterns of transcription factors suspected to be involved in lineage specification would far exceed the capabilities of existing laboratories and technologies that use even large-scale formats such as 96- or 384- well plates.

To overcome this bottleneck, we developed a technology (*Figure 1*) that can massively parallelize the combinatorial transfection of nucleic acids in order to rapidly explore vast search spaces of transcription factor stoichiometries and validated it in an RNA-based screen on human neural progenitor cells (NPCs). By focusing transcription factors onto 1.5 mm spots using dynamically reconfigurable magnetic-field patterns, we significantly reduced the footprint of the experiments to enable screening of transcription factor cocktails over large combinatorial spaces. Importantly, with our technology the amount of RNA and transfection reagent required for a given transcription factor is proportional only to the number of spots actually transfected with that factor, rather than to the total number of possible spots in the array (i.e. to the surface area of transfected cells, not to the total surface area of the plate). This is possible because we focus almost all of the RNA in the medium to the target spots using an array of rare-earth magnets (*Figure 1c*). The number of cell medium changes we use also scale linearly with the number of transcription factors delivered, rather than growing exponentially with the number of combinatorial possibilities. For example, testing every possible combination of just 5 transcription factors results in 31 unique conditions (i.e. all 5 factors alone, 10 possible 2-factor cocktails, 10 possible 3-factor cocktails, 5 possible 4-factor cocktails, and 1 possible 5-factor cocktail) but requires only 5 medium changes when our platform is used.

Standard transfection methods cause considerable fluctuations in the copy numbers of delivered genes, making it impossible to achieve maximal reprogramming efficiency even once optimal transcription factor combinations have been identified. Thus, we also demonstrate here an interleaved RNA transfection technique that dramatically reduces fluctuations in the delivered mRNA copy numbers and stoichiometries. Using our magnetically-guided spotting platform and interleaved transfection protocol, we evaluated the temporal contributions of transcription factor cocktails by treating human NPCs with them during the proliferative stage and/or during the induction of neurogenesis (i.e. after mitogen withdrawal) to generate human dopaminergic neurons with high purity.

## Results

### Modified mRNA constructs and magnetofection

We constructed mRNA expression vectors to address common issues in mRNA delivery and protein expression, such as vector instability, poor translation, and cytotoxicity (*Figure 1a*, see also Materials and methods). The mRNA template was constructed with a T7 promoter and was

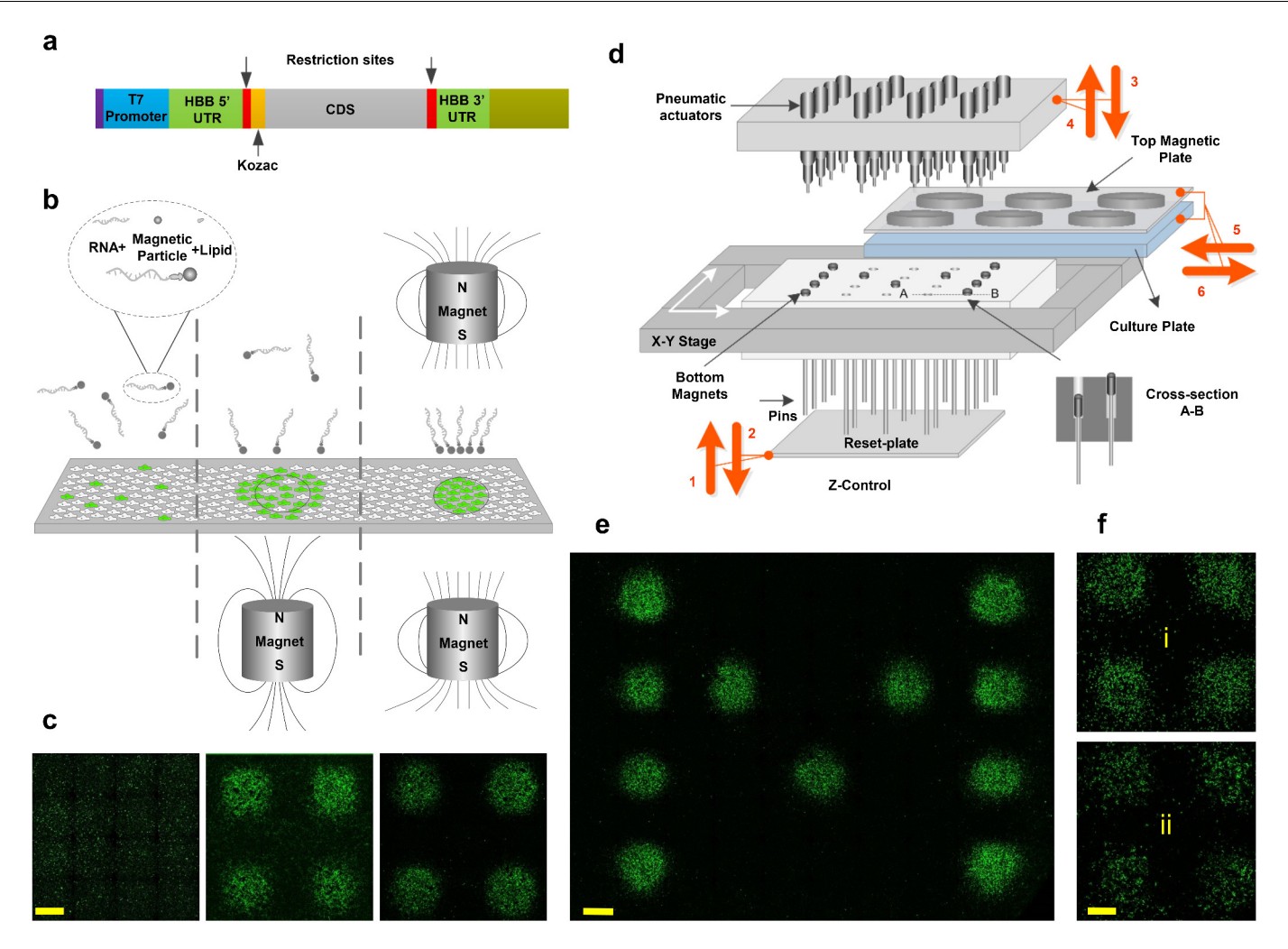

**Figure 1.** Magnetically-guided spotting platform enables localized reconfigurable stoichiometrically-defined mRNA transfection. (a) Schematic of the template used for synthesis of transcription factor mRNAs. (b) Magnetic spotting configurations with single versus dual magnets. (c) Transfection of cultured neural progenitor cells with mRNA encoding green fluorescent protein (GFP) in the absence of a magnetic field (left) or delivered with a single (middle) versus dual (right) magnet setup. (d) Schematic diagram and operation of the automated spotting system. Arrows indicating steps 1 through 6 show the operation order and the movement directions of the system components as described in the text. Steps 1 and 2: Bottom plate resets the positions of bottom magnets to up. Steps 3 and 4: The pneumatic actuators program the positions of bottom magnets by pushing them down. Arrows 5 and 6: The cell-culture plate with top magnets is moved in for transfection and moved out. (e) Demonstration of localized GFP mRNA transfection with user-defined patterns. (f) GFP-transfected differentiating human neural progenitors remain localized the transfected spots at 1 day (i) and 7 days (ii) post-transfection. Scale bars, 1 mm.

DOI: https://doi.org/10.7554/eLife.31922.002

engineered with untranslated regions (UTRs) of highly stable Human β–globin (Hbb) at both 5' and 3' ends (*Angel and Yanik, 2010*), since UTRs play a significant role in mRNA stability (*Yu and Russell, 2001*; *Jiang et al., 2006*). Polyadenylated transcripts with a cap-1 structure were synthesized with complete substitution of uridine and cytidine with the modified nucleosides pseudouridine-5-triphosphate (pseudo-UTP) and 5-methylcytidine-5'-triphosphate (5-methyl-CTP), respectively to increase stability and reduce cytotoxicity by evading the innate immune system (*Karikó et al., 2005*; *Karikó and Weissman, 2007*; *Karikó et al., 2008*; *Stepinski et al., 2001*; *Jemielity et al., 2003*; *Motorin and Helm, 2011*).

Magnetofection has been used for DNA and RNA transfection of cultured cells and yields faster and higher transfection efficiencies than most lipid-based transfection methods, especially for hard-

to-transfect cell types such as primary neurons (*Sapet et al., 2011*; *Plank et al., 2011*). Standard transfection approaches rely on stochastic diffusion and collision of transfection reagents with cells, which is a slow process that transfects cells non-specifically over the entire surface of the cell culture plate (leftmost panels in *Figure 1b* and *Figure 1c*). In magnetofection, transfection vehicles complexed with magnetic nanoparticles (e.g. CombiMag) are drawn towards the target cells using an external magnet positioned underneath the plate. However, due to the non-uniform nature of the magnetic field around a single magnet, magnetofection of cells is not spatially limited by the magnet's physical boundaries and therefore results in partial transfection of cells that lie outside the target area (middle panels in *Figure 1b* and *Figure 1c*). This is a significant limitation for a screening platform intended to precisely restrict transfection to isolated spots using a compact array of small magnets. Magnetotransfection has been traditionally used only in standard bench-top or low-throughput assays.

## Focused delivery of magnetic nanoparticles by dual-magnetic array

To overcome these limitations, we first introduce a simple dual-magnet configuration which focuses the magnetic field to millimeter-size spots in between pairs of magnets positioned above and below the surface of the plate with opposite poles facing one another (rightmost panel in *Figure 1b*). This configuration enables spatially restricted transfection of cells with minimal background transfection outside the boundaries of the magnets (rightmost panels in *Figure 1c*). To make an array of such focused magnetic fields, we note that only the bottom magnets need to be small and spatially defined. Therefore, we use large top magnets that cover the entire wells of the plate (e.g. ~22 mm diameter magnets for a 12-well plate) (*Figure 1d*). This significantly reduces the complexity of setup, requiring us to program the spatial arrangement of only the bottom magnets.

We used neodymium (NdFeB) rare-earth magnets because of their high magnetic field strength compared to other types of magnets. They possess high magnetic anisotropy (i.e. they preferentially align their magnetic moment along an 'easy' axis) and high magnetic coercivity (the ability to resist demagnetisation under an external magnetic field). Both properties are highly desirable in our screening platform, where numerous small magnets are positioned in close proximity. In addition to NdFeB magnets, we tested Alnico five magnets and electromagnets and found both to be inadequate due to insufficient magnetic field (electromagnet; B ~ 10 mT) or failure to withstand demagnetisation (Alnico 5). The top magnetic plate is comprised of 12 large 22 mm diameter NdFeB disk magnets (*Supplementary file 1*) positioned in a 3 × 4 matrix. The precise configuration and dimension of the top magnets is dictated by the desired cell culture plate format (standard 12-well cell culture plates are used in this study). The strength of the magnetic field was measured ~500 mT on the surface of top magnets. The bottom magnetic plate is a matrix of miniature 1.5 mm-diameter NdFeB disk magnets (*Supplementary file 1*) positioned in close proximity (3 mm center-to-center spacing) inside a compact array [an 8 × 12 cm Teflon substrate accommodating a matrix of 25 × 40 (i.e. 1000) magnets]. The magnetic field density on the surface of each magnet is B ~ 300 mT in isolation and B ~ 100 mT when inside the array.

## Programming of the dual-magnet array

In order to drag the mRNA/Lipid/Magnetic particles onto user-defined spot patterns, we program the magnetic field pattern on the substrate by moving each bottom magnet independently closer to ('active') or further away from ('inactive') the bottom of the cell culture plate (*Figure 1d*, see also cross-section A-B there). To achieve this, all bottom magnets are first pushed upwards into the active position by a 'reset-plate' (via the pins mounted underneath each magnet) (*Figure 1d*, Step 1). Afterwards, the reset-plate retracts downwards (*Figure 1d*, Step 2). Since the pins move inside a teflon substrate filled with high viscosity grease, they remain in the elevated active position (5 mm above) after retraction of the reset-plate. Next, selected magnets are pushed down (away from the cell culture plate) using an array of pneumatic actuators located above the magnets (*Figure 1d*, Step 3). This eliminates the magnetic field of these bottom magnets and prevents the transfection of cells at these spots.

The top pneumatic head is comprised of 24 pneumatic actuators assembled in a 6 × 4 matrix. The actuators are sub-miniature stainless steel 5 mm air cylinders (McMaster Carr) with 1 mm diameter piston pins, a 12 mm stroke length, and a spring return mechanism. By activating the actuator of

each piston, the piston pin rapidly pushes the desired magnet down and retracts back. In order to initiate programming, the top pneumatic actuator arm moves down along z-axis until it is in close proximity with bottom magnetic plate (x-y stage sets the initial position of magnetic plate). The pneumatic actuators program (strike) 24 spots simultaneously and then move to the next x-y position. The process automatically proceeds through multiple cycles until all 1000 bottom magnets are programmed. When programming is complete, the top pneumatic actuator arm retracts to its rest position until the next programming event.

Prior to programming the magnets as described above (Steps 1–4 in *Figure 1d*), the cell culture plate is kept off the platform stage in order allow the pneumatic actuators access to the bottom magnets. After programming, the cell-culture plate containing with the desired mRNA complex, along with the top magnets, is moved into the platform for transfection and afterwards moved out (Steps 5 and six in *Figure 1d*). Steps 1–6 are then repeated using a new mRNA complex until all transcription factors have been delivered to the specified spots. mRNA transfection of human NPCs under a user-defined pattern is shown in *Figure 1e*. With respect to cell mobility and cell-cell interactions, the spatial stability of the spots post-transfection and during differentiation could be of concern. However, *Figure 1f* shows that the GFP-transfected spots of proliferating and differentiating NPCs remain isolated from each other even after 7 days post-transfection and our results in the subsequent sections show the stability of differentiated cell spots even after 17 days (i.e. when the differentiation protocol is completed).

By focally localizing all transfection components (i.e. mRNA, lipid-based transfection reagent, and magnetic particles) onto a very small footprint, our magnetically-guided spotting platform achieves a significant savings in reagents relative to 96-well plate screens. Unless otherwise noted, we deliver 0.6 ng of mRNA per 1.5 mm diameter spot (total surface area per spot: ~1.8 mm$^2$) for all of our experiments. Achieving a comparable level of transfection in a 96-well format (total surface area per well: ~32.2 mm$^2$) requires ~18 fold more reagents. Although the reagent requirements for multiwell plate screens can be further reduced with 384- or 1536-well plate assays, these smaller formats suffer significantly from variation due to edge effects, which also cause problems for even 96-well screens (*Lundholt et al., 2003*). Our magnetically-guided spotting platform offers the space and reagent savings characteristic of small multiwell assays in larger-format plates, thus ensuring highly uniform culture conditions for accurate comparison of all combinatorial conditions. Uniformity of delivered RNA copy-numbers and stoichiometries

The fact that cell fate decisions are highly dependent on the precise expression levels of a limited set of genes demands that each reprogramming factor in a given cocktail needs to be delivered with minimal cell-to-cell variation to achieve maximal efficiency. Variable expression and low overall efficiency remain major drawbacks of existing DNA transfection protocols, particularly when working with difficult to transfect cell types such as primary and postmitotic cells (*Hansson et al., 2015*; *Landi et al., 2007*). DNA-based delivery methods are most vulnerable to low/variable efficiency because transfected molecules must be transcribed in order to be effective and therefore must cross both the plasma membrane and the nuclear envelope. In contrast, mRNA-based methods tend to yield much higher transfection efficiencies since transfected molecules need to cross only the plasma membrane to be expressed. For example, a comparison of plasmid- and mRNA-based transfection approaches in human embryonic stem cell-derived retinal pigmented epithelial cells shows that DNA efficiency is only ~10% while RNA efficiency is ~90% (*Hansson et al., 2015*). Similarly, an extensive comparison of DNA and RNA transfection protocols using immature dendritic cells suggests that DNA-based approaches can achieve transfection efficiencies of 10–20% whereas RNA-based approaches can achieve efficiencies of 40–80% (*Landi et al., 2007*). Due to the low expression efficiency of DNA-based transfection methods, the stochastic noise in the expression is large. Higher expression results in correspondingly lower cell-to-cell variation as the Poisson distribution approaches a normal distribution. This is supported by the low variability of RNA transfection results with respect to DNA transfection (*Hansson et al., 2015*; *Landi et al., 2007*).

Here, we use our magnetically-guided spotting platform to show that we are able to further reduce variation in the copy number of transfected mRNAs by breaking the transfection process into multiple temporally-segregated transfections, where each transfection is done using a reduced concentration of the transfection complex (*Figure 2a*). Human neural progenitor cells were cultured in 12-well plates and spotted with mRNAs encoding GFP (Green) or mCherry (Red) using either a standard single transfection process (32 ng of each mRNA) or a quadruple transfection protocol (four

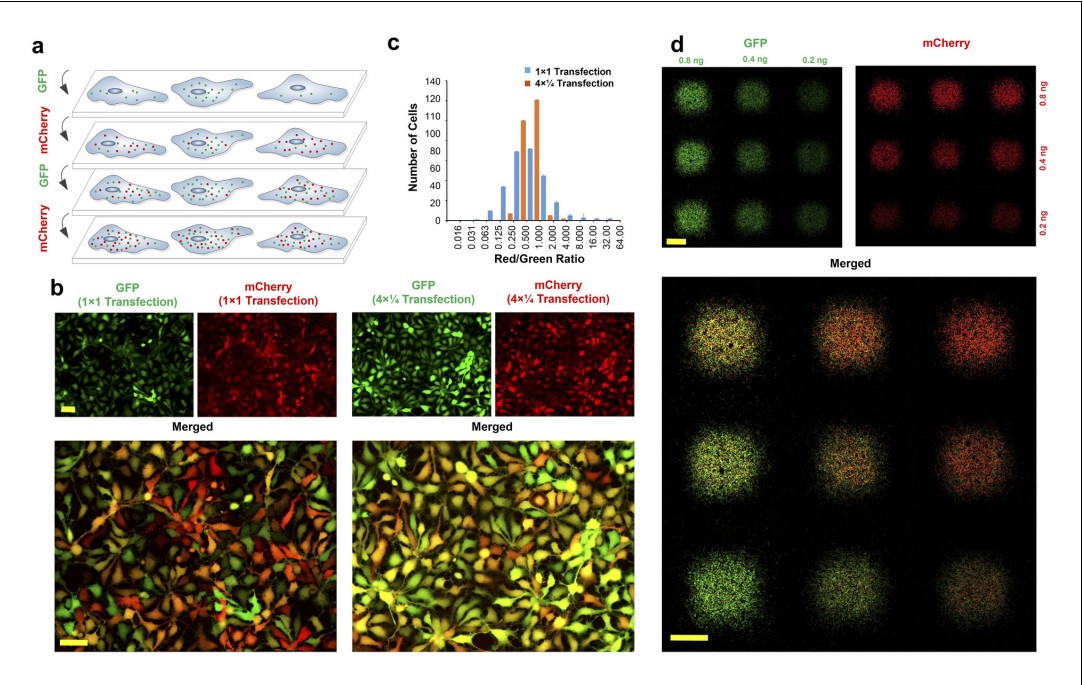

**Figure 2.** Interleaved transfection reduces fluctuations in delivered mRNA copy numbers and stoichiometries. (**a**) Schematic illustration of multiple interleaved transfections using *GFP* and *mCherry* mRNAs. Each mRNA is delivered in multiple interleaved rounds of transfection, as opposed to being delivered all at once in a single transfection. (**b**) Neural progenitor cells transfected with *GFP* and *mCherry* mRNA using a standard single transfection protocol ('1 × 1'; i.e. the full dose of each mRNA is applied in a single transfection) versus an interleaved transfection protocol ('4 × ¼"; i.e. one quarter doses of each mRNA are applied using four interleaved transfections). All images were acquired using automated software to prevent saturation. (**c**) Ratio of red (mCherry) to green (GFP) fluorescence intensity per cell for both single and interleaved transfection protocols. Image correlation analysis was performed on the images shown in (**b**) using WCIF plugin for ImageJ. Efficiency of interleaved transfection is calculated using Pearson's Correlation Coefficient (PCC) and Mander's Overlap Coefficient (MOC). PCC = 0.85 and MOC = 0.92 for interleaved transfections compared to PCC = 0.54 and MOC = 0.79 for single transfections (−1 < PCC < 1; 0 < MOC < 1). Both PCC and MOC represent average values from three independent experiments. (**d**) Precise RNA dosage control using the magnetically reconfigurable spotting platform. *GFP* mRNA was spotted in a 3 × 3 matrix at three different dosages: right column (1x), middle column (2x) and left column (4x). *mCherry* mRNA was delivered to the same spots at three different dosages as follows: bottom row (1x), middle row (2x) and top row (4x). To generate this pattern, we delivered *GFP* and *mCherry* mRNAs using our interleaved transfection protocol to achieve highest co-transfection efficiency (this pattern is also reproducible using standard transfection protocols, although with reduced efficiency). In the process, 1x concentration of *GFP* (1.8 ng; 0.2 ng per spot) was delivered to the plate while all nine magnets were active. In the next step, another 1x concentration of *GFP* (1.2 ng; 0.2 ng per spot) was delivered to the cells while the 3 magnets of the rightmost column were inactive. Finally, 2x concentration of *GFP* (1.2 ng; 0.4 ng per spot) was delivered to the cells while the middle and the rightmost columns of 6 magnets were inactive. A similar process was repeated for the *mCherry* along the perpendicular direction (i.e. by activating/inactivating magnets along the horizontal rows rather than the vertical columns). Scale bars: 100 µm in (**b**), 1 mm in (**d**).

DOI: https://doi.org/10.7554/eLife.31922.003

The following source data is available for figure 2:

**Source data 1.** Ratio of red (mCherry) to green (GFP) fluorescence intensity per cell for both single and interleaved transfection protocols.
DOI: https://doi.org/10.7554/eLife.31922.004

transfections consisting of 8 ng of each mRNA per transfection). In both cases the total amount of mRNA delivered to the spots was 64 ng and the medium was changed after each transfection. As shown in *Figure 2b*, although both methods result in a majority of all cells being transfected, cell-to-cell expression is considerably more variable when using the single-transfection method. To quantify co-localization efficiency, we compared the ratio of mCherry vs. GFP fluorescence intensity for each cell following single-step and multi-step ('interleaved') transfection protocols using the Pearson correlation coefficient (PCC) and Mander's overlap coefficient (MOC) (*Figure 2c*) (*Li et al., 2004*; *Manders et al., 1993*). Although a single transfection results in co-localization (PCC >0.5) with 79% overlap (MOC = 0.79), interleaved transfection improves the PCC to 0.85 and MOC to 0.92, which is indicative of highly uniform and co-localized transfection of cells with both mRNAs. While the interleaved protocol results in superior co-expression efficiency (i.e. transfection uniformity across all

cells), the single-transfection method is more than adequate for large-scale screens where the priority is to compare the largest number of transcription factor combinations head-to-head under identical culture conditions while simultaneously minimizing the number of medium changes. The best performing mRNA combinations can then be retested using the interleaved protocol to precisely quantify their maximum efficiency.

Precise dosage control and transfection gradients can be achieved as shown in *Figure 2d* where NPCs were spotted with *GFP* and *mCherry* mRNAs at three different dosages horizontally and vertically, respectively. This resulted in a matrix of spots with varying combinatorial dosages of two mRNAs ranging from 4x/4x (top left spot) to 1x/1x (bottom right spot).

## Screening of transcription factors for dopaminergic neurons

Parkinson's disease is one of the most common neurodegenerative disorders resulting from the functional loss of dopaminergic neurons in *substantia nigra pars compacta* of midbrain (*Lees et al., 2009*). Dopamine replacement therapy and deep-brain stimulation can improve the quality of life, however the long-term shortcomings of these treatments make the alternative option of cellular replacement attractive. Therefore, deriving dopaminergic neurons in vitro from pluripotent stem cells or progenitors either for cell therapy or for basic research is of critical importance, since other cell resources are quite limited and unreliable. However, deriving sufficiently pure cultures of dopaminergic neurons remains a key challenge.

To directly assess the contributions of many exogenous gene expression cocktails on the efficiency of dopaminergic neuron differentiation, we performed a combinatorial screen of transcription factors delivered by modified mRNA. We used proliferative NPCs, due to their relative ease to expand and culture as monolayers in the presence of growth factor mitogens (bFGF and EGF), and conducted the screen using a pool of midbrain-specific transcription factors reported in the literature to be involved in either the fate specification, survival, or maintenance of ventral midbrain dopaminergic cells (*OTX2, LMX1A, FOXA2, ASCL1, NGN2, NURR1,* and *PITX3*) (*Hegarty et al., 2013*; *Abeliovich and Hammond, 2007*; *Ang, 2006*). While *OTX2, ASCL1,* and *NGN2* are essential during early neural development for the correct positioning of isthmus organizer and the development of the midbrain (*Martinez-Barbera et al., 2001*; *Puelles et al., 2003*; *Kele et al., 2006*), *FOXA2* and *LMX1A* are necessary for the correct positioning of dopaminergic cell types along the ventral-caudal axis of the midbrain (*Nakatani et al., 2010*). *PITX3* has been shown to be expressed in all dopaminergic neurons in the CNS during maturation and, together with NURR1, is necessary for their survival and maintenance (*Saucedo-Cardenas et al., 1998*; *Peng et al., 2011*; *Martinat et al., 2006*). Based on these findings, we hypothesized that temporally controlled delivery of these transcription factor combinations might be important for dopaminergic neuron differentiation.

We used two different mediums: (1) an initial expansion medium containing the mitogens bFGF and EGF (proliferative NPC stage) and (2) a subsequent differentiation induction medium lacking the mitogens in order to induce neurogenesis (induction stage; *Figure 3a*). On day 4, the differentiation medium was supplemented with BDNF, GDNF, and ascorbic acid to improve cell survival, and also with dibutyryl-cAmp and TGF-β3 to enhance differentiation. Neural differentiation media formulated with these supplements have been shown to support dopaminergic differentiation (*Kriks et al., 2011*; *Maroof et al., 2013*).

Transcription factors were delivered individually either during the proliferative NPC stage (Day −2, before mitogen removal) or the induction stage (Day 0, upon mitogen removal) as indicated by the superscripts N or I on each factor, respectively. Immunofluorescence results at day 17 indicated a significant increase in tyrosine hydroxylase (TH)$^+$ neuron yield only in spots transfected with *LMX1A, FOXA2,* or *PITX3* (*Figure 3b and d*). Interestingly, among spots transfected during induction stage, *LMX1A* (Lm$^I$) was the only factor that significantly increased the yield of TH$^+$ neurons on its own (p<0.01, 40 ± 5% TH$^+$/MAP2$^+$ cells compared to the control, 22 ± 4%). On the other hand, significantly more TH$^+$ neurons were counted in spots transfected during the NPC stage with either *FOXA2* (Fo$^N$) (TH$^+$/MAP2$^+$, 27 ± 4%, p<0.05), *PITX3* (Pt$^N$) (TH$^+$/MAP2$^+$, 35 ± 3%, p<0.01), or *LMX1A* (Lm$^N$) (TH$^+$/MAP2$^+$, 32 ± 3%, p<0.01), albeit with less effect than post-induction transfection with *LMX1A* (Lm$^I$) (*Figure 3b*). These results suggest that *FOXA2* and *PITX3* are more efficient when delivered to NPCs whereas *LMX1A* is more efficient when delivered during induction of differentiation after mitogen removal. Although *OTX2, ASCL1, NGN2* are thought to play a role in dopaminergic differentiation, in our experiment their direct overexpression did not increase TH$^+$ neurons. Since

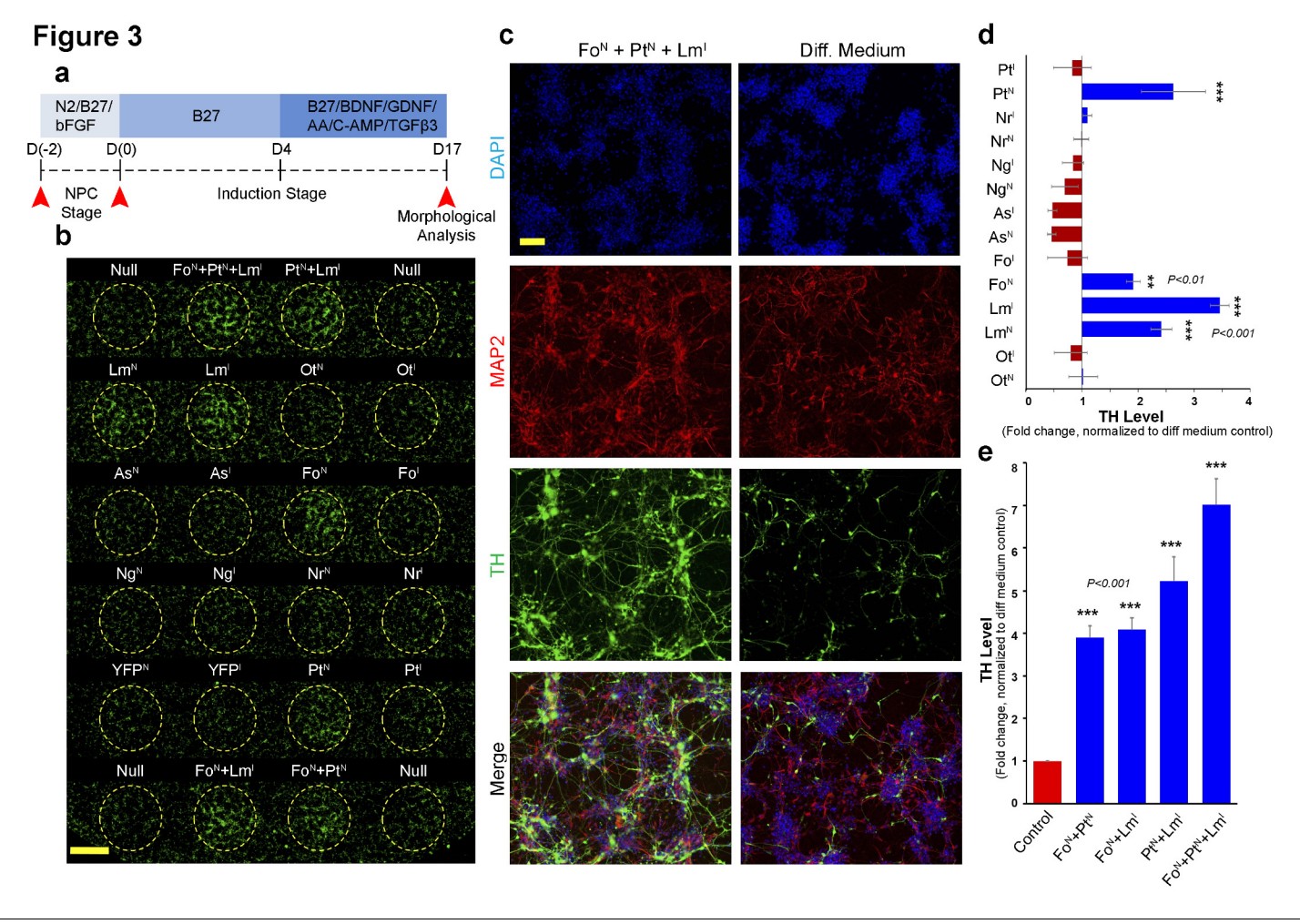

**Figure 3.** Magnetically reconfigurable spotting platform can screen and identify key transcription factors promoting dopaminergic cell fate. (a) Timeline of the mRNA-induced differentiation process (proliferative NPC stage from day −2 to 0; early induction stage after day 0). Red arrows indicate time points of transfection before mitogen growth factor removal, transfection after growth factor removal (induction stage), and the final analysis at 17 days after growth factor removal. (b) Immunocytochemistry at day 17 for TH (green) showing upregulation of TH when *FOXA2* is delivered to NPCs (Fo$^N$), when *PITX3* is delivered to NPCs (Pt$^N$), and when *LMX1A* is delivered either to NPCs (Lm$^N$) or during induction (Lm$^I$). Combinations of these three conditions also increase the number TH$^+$ neurons. (c) Double staining (MAP2$^+$/TH$^+$) results from cells transfected with the most effective combination (Fo$^N$ +Pt$^N$ + Lm$^I$) compared to non-transfected cells (Diff. Medium). (d and e) Quantitative TH gene expression analysis at day 17 comparing all single factors delivered at NPC ($^N$) and early induction ($^I$) stages (d), and temporal transfection of the selected combinations of double and triple factors (e). All quantifications were done with n = 3 independent experiments (mean ±s.e.m), ***($p<0.001$), **($p<0.01$), *($p<0.05$) (compared to control, Dunnett's test). Abbreviations: TH, tyrosine hydroxylase; Ot, *OTX2*; Lm, *LMX1A*, Fo, *FOXA2*; As, *ASCL1*; Ng, *NGN2*; Nr, *NURR1*; Pt, *PITX3*; Superscript '$^N$', proliferative NPC stage; Superscript '$^I$', induction stage after mitogen withdrawal. Scale bars: 1 mm in (b), 100 μm in (c).

DOI: https://doi.org/10.7554/eLife.31922.005

The following source data is available for figure 3:

**Source data 1.** Quantitative TH gene expression analysis at day 17 comparing factors delivered at NPC (superscript N) and early induction (superscript I) stages.

DOI: https://doi.org/10.7554/eLife.31922.006

we used human NPCs, previous results reporting that *ASCL1* enhances TH +neurons derived from IPSCs (*Theka et al., 2013*)may indicate that *ASCL1* is only required during pre- or immediate early-neurogenesis stages. *OTX2* expression may also have a similar role, since it is known to be necessary for the early patterning of midbrain/hindbrain regional identity.

We next investigated the temporal requirement of the selected transcription factors using combinatorial screening (Fo$^N$, Pt$^N$ and Lm$^I$). While *FOXA2* (Fo$^N$) and *PITX3* (Pt$^N$) were only used for pre-

induction NPCs, post-induction was performed only with *LMX1A* (Lm$^I$) on differentiating cells. Although all dual combinations increased the number of converted cells, triple combination of these factors (Fo$^N$ +Pt$^N$ + Lm$^I$) produced significantly more TH$^+$ cells (p<0.001, 68 ± 5% TH$^+$/MAP2$^+$) than any other combination (*Figure 3c*). Gene expression analysis at day 17 of differentiation (*Figure 3e*) confirms robust seven-fold over-expression of TH in response to the Fo$^N$ +Pt$^N$ + Lm$^I$ triple combination of factors when compared to non-transfected cells.

We then proceeded to characterize the identity of the TH$^+$ cells derived from the Fo$^N$ +Pt$^N$ + Lm$^I$ combination in greater detail using gene expression analysis for selected mid-brain specific neuronal makers. We used standard lipid-based transfections (Stemfect reagent; 12-well microplates) for this experiment and performed parallel gene expression and immunocytochemical analysis (staining for DAPI, MAP2, FOXA2 and TH) (*Figure 4a and b*; a list of TaqMan assays used in this research is given in *Supplementary file 2*). Neuronal identity (MAP2$^+$) was observed in ~98% of both Fo$^N$ +Pt$^N$ + Lm$^I$ transfected cells and the non-transfected controls. Double staining indicated robust expression of FOXA2 in both transfected (p<0.05, 95 ± 2%, FOXA2$^+$/DAPI$^+$) and non-transfected (70 ± 4%, FOXA2$^+$/DAPI$^+$) cells, suggesting the differentiated cells in both populations adopted a ventral mid-brain identity (*Figure 4b*). However, transfected cells yielded significantly higher FOXA2$^+$ neurons co-expressing TH (p<0.001, 75 ± 4%, TH$^+$/FOXA2$^+$) compared to non-transfected cells (30 ± 3% FOXA2$^+$/TH$^+$). Gene expression analysis also confirmed the robust up-regulation of numerous mid-brain-specific markers compared to non-transfected controls (*Figure 4c*). Among the transfected transcription factors, *LMX1A* was up-regulated by eleven-fold, *FOXA2* by nine-fold and *PITX3* by five-fold 2 weeks after transfection with the Fo$^N$ +Pt$^N$ + Lm$^I$ combination. Surprisingly, two of the non-effective transcription factors used in our initial screen (*OTX2* and *NURR1*) were also up-regulated by eight- and four-fold respectively. A well-known midbrain regional marker expressed in *substantia nigra* DA neuron sub-types (*VMAT2*) was also up-regulated by more than five-fold, further confirming that our optimized triple combination of transcription factors (Fo$^N$ +Pt$^N$ + Lm$^I$) gives rise to dopaminergic neurons with midbrain identity.

## Discussion

During the past decade, there have been significant efforts to generate different cell types, such as dopaminergic neurons, from human pluripotent stem cells (hPSCs) (*Kriks et al., 2011*; *Theka et al., 2013*; *Friling et al., 2009*; *Lee et al., 2010*), typically by supplementing the cell medium with various growth factors. In addition to growth factor-based differentiation protocols, exogenous overexpression of transcription factors (such as the single transcription factor *LMX1A* [*Friling et al., 2009*] or the combined overexpression of *FOXA2* and *NURR1* [*Lee et al., 2010*]) using DNA vectors can partially enhance the number of TH positive dopaminergic neurons during differentiation of pluripotent stem cells under various culture conditions. However, directly comparing differentiation efficiencies across laboratories has been impractical due to the differences in experimental protocols. Although various yields and purities of dopaminergic neural differentiation from hPSCs and/or mutipotent NPCs have been reported—ranging from low (i.e. 10%) to high (80%)—much of this variation can be attributed to differences in starting cell lines, growth factor cocktails, use of recombinant vs. purified proteins, cell purification procedures, and/or co-culture with other cells such as astrocytes (*Engel et al., 2016*). Our high-throughput magnetically-guided spotting platform, where many factors and conditions can be simultaneously tested side-by-side and compared under the same laboratory conditions, can in the future allow for more accurate and quantitative comparisons.

Our high-throughput technology enables the rapid exploration of large combinatorial spaces of transcription factors by massively parallelizing the uniform delivery and transfection of mRNAs to unusually small spot sizes. In the present study, we demonstrate the combinatorial and temporal delivery of a pool of midbrain-specific transcription factors to generate dopaminergic neurons. We show that combinatorial delivery of *LMX1A*, *FOXA2*, and *PITX3* is highly effective in generating dopaminergic neurons from neural progenitors, with *LMX1A* significantly increasing TH-expression levels when delivered during both the proliferating NPC stage and in the early neural induction stages upon mitogen withdrawal, while *FOXA2* and *PITX3* only exhibit high efficacy before neural induction.

## Figure 4

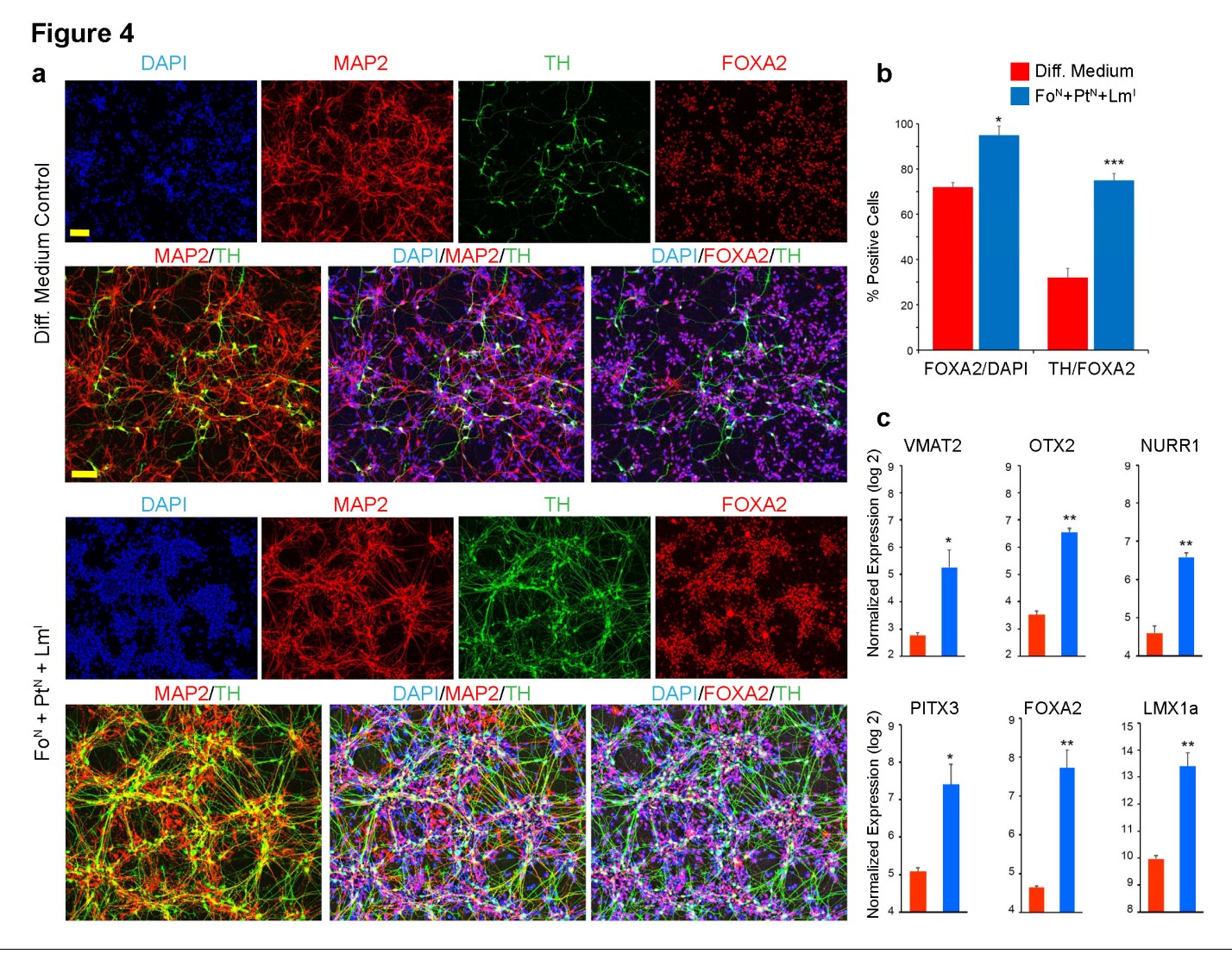

**Figure 4.** Immunocytochemical and quantitative analysis confirms midbrain identity dopaminergic neurons generated from screen. (a) Immunocytochemistry at day 17 for TH (green), MAP2 (red), FOXA2 (red) and DAPI (blue) performed in microwell plates; comparing triple staining DAPI$^+$/MAP2$^+$/TH$^+$ and DAPI$^+$/FOXA2$^+$/TH$^+$ between non-transfected cells in differentiation medium alone (top) to Fo$^N$ + Pt$^N$ + Lm$^I$ transfected cells (bottom). (b) Quantification of data presented in (a). The percentage of cells expressing FOXA2 increases from 72% in non-transfected cells to 95% in cells transfected with the combination of Fo$^N$ + Pt$^N$ + Lm$^I$ (left panel, FOXA2/DAPI). The percentage of the *FOXA2* positive cells that also co-express TH increases from 32% in non-transfected cells to 75% in cells transfected with the combination of Fo$^N$ + Pt$^N$ + Lm$^I$ (right panel, TH/FOXA2). (c) Gene expression analysis at day 17 for the indicated midbrain specific markers. Quantifications were done with n = 3 independent experiments (mean ± s.e. m), ***p<0.001, **p<0.01, *p<0.05 (compared to control, Student's *t*-test). Abbreviations: TH, tyrosine hydroxylase; Lm, *LMX1A*; Fo, *FOXA2*; Pt, *PITX3*; Superscript '$^N$', proliferative NPC stage; Superscript '$^I$', induction stage after mitogen withdrawal. Scale bars, 100 μm.

DOI: https://doi.org/10.7554/eLife.31922.007

# Materials and methods

## Key resources table

| Reagent type (species) or resource | Designation | Source or reference | Identifiers |
|---|---|---|---|
| Gene (*Homo sapiens*) | OTX2 | NA | HGNC:8522 |
| Gene (*H. sapiens*) | LMX1A | NA | HGNC:6653 |

*Continued on next page*

*Continued*

| Reagent type (species) or resource | Designation | Source or reference | Identifiers |
|---|---|---|---|
| Gene (*H. sapiens*) | FOXA2 | NA | HGNC:5022 |
| Gene (*H. sapiens*) | ASCL1 | NA | HGNC:738 |
| Gene (*H. sapiens*) | NGN2 | NA | HGNC:13805 |
| Gene (*H. sapiens*) | NURR1 | NA | HGNC:7981 |
| Gene (*H. sapiens*) | PITX3 | NA | HGNC:9006 |
| Cell line (*H. sapiens*) | hNP1 | EMD Millipore | EMD Millipore: SCR055; RRID:CVCL_GS51 |
| Antibody | anti-FOXA2 (mouse monoclonal) | Abcam | Abcam:ab60721 |
| Antibody | anti-MAP2 (chicken polyclonal) | Abcam | Abcam:ab5392 |
| Antibody | anti-Tyrosine Hydroxylase (rabbit polyclonal) | Abcam | Abcam:ab112 |
| Sequence-based reagent | VMAT2 (TaqMan assay) | Thermo Fisher Scientific | Thermo Fisher Scientific: Hs00996835_m1 |
| Sequence-based reagent | GAPDH (TaqMan assay) | Thermo Fisher Scientific | Thermo Fisher Scientific: Hs02758991_g1 |
| Sequence-based reagent | OTX2 (TaqMan assay) | Thermo Fisher Scientific | Thermo Fisher Scientific: Hs00222238_m1 |
| Sequence-based reagent | PITX3 (TaqMan assay) | Thermo Fisher Scientific | Thermo Fisher Scientific: Hs01013935_g1 |
| Sequence-based reagent | FOXA2 (TaqMan assay) | Thermo Fisher Scientific | Thermo Fisher Scientific: HS00232764_m1 |
| Sequence-based reagent | LMX1A (TaqMan assay) | Thermo Fisher Scientific | Thermo Fisher Scientific: Hs00892663_m1 |
| Sequence-based reagent | TH (TaqMan assay) | Thermo Fisher Scientific | Thermo Fisher Scientific: Hs0016594_m1 |
| Sequence-based reagent | MAP2 (TaqMan assay) | Thermo Fisher Scientific | Thermo Fisher Scientific: Hs00258900_m1 |
| Sequence-based reagent | NURR1 (TaqMan assay) | Thermo Fisher Scientific | Thermo Fisher Scientific: Hs00443062_g1 |
| Other | DAPI stain | Thermo Fisher Scientific | |

## Cell culture

The H9-derived human neural progenitor cell line (hNP1) was obtained from Aruna Biomedical (Athens, GA; now distributed as ENStem-A, EMD Millipore #SCR055). Each lot of ENStem-ATM Human Neural Progenitor Cells has been validated for high levels of expression of Nestin and Sox2 and low level expression of Oct4. The ability of ENStem-ATM cells to differentiate into multiple neuronal phenotypes and maintain a normal karyotype after multiple passages has been verified by the manufacturer, and the cells have been confirmed to be negative for mycoplasma. Cells were expanded on Matrigel-coated (Corning Inc., Corning, NY) six-well plates with the supplier's expansion medium. Starting from passage two, the expansion medium was gradually changed (25% medium replacement per passage) to N2/B27 in DMEM/F12 supplied with bFGF/EGF (20 ng/ml,

Thermo Fisher Scientific, Waltham, MA). Cells were passaged at the split ratio of 1:2 by cell scraper. All experiments were done with cells in passages 7–11.

## Synthesis of modified synthetic transcription factor mRNAs

Our mRNA synthesis methodology has been previously described (*Angel and Yanik, 2010*). Briefly, dsDNA templates were linearized from cDNA clones in pCMV6 vectors for *OTX2*, *LMX1A*, *FOXA2*, *ASCL1*, *NGN2*, *NURR1*, *PITX3*, *EGFP*, and *mCherry* (OriGene, Rockville, MD). To maximize the stability of mRNA transcripts and increase protein translation, we added 5' and 3' untranslated regions from human Beta-globin gene (Hbb) to the templates by ligating with *E. coli* DNA ligase (New England Biolabs, Ipswich, MA). A T7 promoter was also added to the 5' UTR to facilitate in vitro transcription. Assembled templates were cloned into the pCR-Blunt II TOPO vector and inserted in TOP10 chemically competent *E Coli* to propagate using the Zero Blunt TOPO PCR Cloning Kit (Thermo Fisher Scientific). Plasmids were purified using EndoFree Plasmid Maxi Kit (Qiagen, Hilden, Germany). Templates were linearized from plasmids then amplified by high-fidelity PCR using KAPA HiFi PCR Kit (KAPA Biosystems, Wilmington, MA). PCR products were separated by agarose gel electrophoresis and purified using QIAquick Gel Extraction Kit (Qiagen). Cap 1-capped, poly(A) tailed mRNA was synthesized from the purified templates by in vitro transcription using the mScript mRNA Production Kit (CellScript, Madison, WI). All mRNAs were synthesized with complete substitution of uridine and cytidine with the modified nucleosides pseudouridine-5-triphosphate (pseudo-UTP) and 5-methylcytidine-5'-triphosphate (5-methyl-CTP), respectively. The mRNAs were purified using an RNeasy Mini kit (Qiagen). To ensure the transcripts were produced with right poly(A) tail, we analysed the samples both before and after tailing using formaldehyde-agarose gel electrophoresis. SUPERase In RNase Inhibitor (Thermo Fisher Scientific) was added to mRNAs at concentration of 1 ug/20 ug.

## RNA transfection and differentiation

For directed differentiation experiments without spotting, cells from passages 7–11 were plated on 12-well plates in expansion medium (DMEM/F12/N2/B27/bFGF/EGF). For temporal analysis, growth factors (bFGF and EGF) were kept in the medium for two additional days, which led us to seed the cells at the appropriate split ratio to reach the optimal surface confluency of 30–40% the next day and final cell density of 90% after removal of growth factors. For mRNA transfection, we used lipid-based Stemfect RNA transfection kit (Stemgent, Cambridge, MA) and complexed it with each RNA separately with 10 min of incubation. For all transfections, 100 ng total RNA was complexed with 1 µl Stemfect reagent in 20 µl Stemfect buffer. Cell culture medium was changed to N2/B27 without growth factors, and complexed RNA was resuspended in the medium followed by 4 hr of incubation. After transfection, for transfection at the proliferative NPCs stage ($^N$), the medium was changed back to N2/B27/bFGF/EGF for two more days. For transfection post-induction ($^I$), the medium was changed to B27 without growth factors 4 hr prior to transfection, and followed by the addition of RNA/lipid complex, 4 hr of incubation, and changing to fresh B27 medium. For both pre- and post-neural induction (with or without mitogens, respectively) transfection protocols, differentiation was started 72 hr after plating the cells by removing the growth factors and changing the medium to B27. The medium was changed every other day. On day 4, medium was supplemented with brain-derived neurotrophic factor (BDNF, 20 ng/ml; R and D Systems, Minneapolis, MN), glial-derived neurotrophic factor (GDNF, 20 ng/ml; R and D Systems), ascorbic acid (0.2 mM; Tocris), dibutyryl cyclic adenosine monophosphate (Dc-Amp, 0.5 mM; Sigma-Aldrich, St. Louis, MO), and transforming growth factor type β3 (TGF-β3, 1 ng/ml; R and D Systems). Cells were cultured in this medium until they were ready for end-point assays at day 17 post-differentiation, and were either fixed for image analysis or collected for qRT-PCR analysis.

## Magnetic spotting methodology

For magnetofection of cells with spotted synthetic mRNA, cells from passages 7–11 were transferred to 12-well glass-bottom plates (Cellvis, Mountain View, CA) at optimised density of 30–40% in expansion medium (DMEM/F12/N2/B27/bFGF/EGF). With the 12-well plate format, a total number of 36 magnetic hotspots could be accommodated under each well. In order to deliver the RNA to the magnetic hotspots, we used CombiMag magnetofection transfection reagent (OZBiosciences,

Marseille, France). CombiMag contains magnetic nanoparticles that can be mixed with all transfection reagents and improves their efficiency by means of magnetic field. We mixed mRNA with magnetic nanoparticles and complexed it with Stemfect reagent during a 5–10 min incubation according to the manufacturers' recommended ratios. Localized transfection was initiated by replacing the medium in the wells with N2/B27 medium in which the complexed mRNA/lipid/nanoparticles had been resuspended. The concentration of the mRNA complex was adjusted based on the number of spots to be transfected per well (0.6 ng RNA per 1.5 mm diameter spot), ensuring that each spot received an equivalent amount of mRNA. Plates were then moved onto the spotting system with the desired magnetic patterns pre-programmed as described. The incubation time for magnetofection was 2 min per transfection. This transfection process was repeated for each mRNA factor until all desired combination were delivered to the spots. The medium was changed between each transfection with new N2/B27 containing the appropriate mRNA complex. The number of medium changes scale linearly with the number of transcription factors delivered to each well. For temporal transfection during the proliferative NPC stage, growth factors were added to the medium after the final round of spotting. For differentiation, the medium was changed to B27 with the remainder of the protocol similar to the standard differentiation protocols discussed in the previous section.

## Immunocytochemistry

Cell were fixed in 4% paraformaldehyde in TBST for 15 min, permeabilized for 10 min in 0.2% Triton X-100, blocked with 1% casein for 60 min, and incubated overnight at 4°C with appropriate antibodies in 50/50 1% casein/TBST. Fixed cells were then incubated with secondary antibodies for 60 min (Alexa Fluor 488, 555 and 647 Dye, Molecular Probes, Eugene, OR). DAPI (Thermo Fisher Scientific) was used as nuclear counterstain. Antibodies used are as follows: mouse Foxa2 (ab60721, Abcam, Cambridge, MA), chicken MAP2 (ab5392, Abcam) and rabbit TH (ab112, Abcam)

## Gene expression analysis (qRT-PCR)

Total RNA was collected and purified using an RNeasy Mini Kit (Qiagen), measured using a Nano-Drop 1000 spectrophotometer (Thermo Fisher Scientific), and gene expression analysis was performed using commercially available TaqMan gene expression assay (Applied Biosystems, Foster City, CA; a list of TaqMan assays is given in *Supplementary file 2*). The qRT-PCR was done in one-step, 20 µl reactions with 15 min reverse transcription at 50°C, 2 min initial denaturing at 95°C followed by 40 cycles of 15 s/95°C and 1 min/60°C. Three individual samples with three replicates each were used for gene expression analysis, and the data were normalized to *GAPDH*.

## Cell counting and statistical analysis

For image acquisition, we used a high-performance laser-based confocal imaging system (INCell 6000, GE Healthcare, Chicago, IL). For non-spotting experiments, a total of 27 random images were taken for each condition from three independent experiments. For spotting experiments, images were taken from spotted areas in three independent experiments. Cell counting was performed using CellProfiler (*Carpenter et al., 2006*). To count the cells, we first counted the number of DAPI-positive cells, followed by counting the number of cells expressing the marker of interest. Student's *t*-test (comparing two groups) was used for statistical analysis. Co-localization analysis was performed using the WCIF-ImageJ software package and the Image Correlation Analysis plugin (*Schneider et al., 2012*). The plugin uses Pearson's Correlation Coefficient (PCC) for quantifying the correlation between two channels, as well as calculating Mander's Overlap Coefficient (MOC).

## Code availability

LabView files for programming of the dual-magnet array are available online from GitHub (https://github.com/rezaie99/ELIFE-050518; copy archived at https://github.com/elifesciences-publications/Yanik_et_al_2018) (*Ghannad-Rezaie, 2018*).

## Acknowledgement

This project was funded by NIH Director's Pioneer Award (DP1 OD006782) and Packard Award for Science and Engineering.

## Additional information

### Funding

| Funder | Grant reference number | Author |
|---|---|---|
| NIH Office of the Director | NIH Director's Pioneer Award 1DP1OD006782 | Mehmet Fatih Yanik |
| David and Lucile Packard Foundation | Packard Fellowship for Science and Engineering | Mehmet Fatih Yanik |

The funders had no role in study design, data collection and interpretation, or the decision to submit the work for publication.

### Author contributions

Sayyed M Azimi, Data curation, Software, Formal analysis, Validation, Investigation, Visualization, Methodology, Writing—original draft, Writing—review and editing, Designed and performed the experiments, Analyzed the results, Was involved in the writing of the manuscript; Steven D Sheridan, Data curation, Formal analysis, Validation, Investigation, Visualization, Methodology, Writing—original draft, Provided advice and training to SM Azimi, Participated in the design and analysis of the experiments, Was involved in the writing of the manuscript; Mostafa Ghannad-Rezaie, Methodology, Writing—original draft, Writing—review and editing, Developed the spotting hardware with SM Azimi, Was involved in the writing of the manuscript; Peter M Eimon, Supervision, Methodology, Writing—review and editing, Worked on developing RNA-mediated differentiation protocols and on editing and preparing the manuscript for publication; Mehmet Fatih Yanik, Conceptualization, Resources, Supervision, Funding acquisition, Investigation, Writing—original draft, Project administration, Writing—review and editing, Was the Principle Investigator, Conceived the study, Worked on analysis of the experiments, and was involved in the writing of the manuscript

### Author ORCIDs

Peter M Eimon (iD) https://orcid.org/0000-0003-0447-517X
Mehmet Fatih Yanik (iD) https://orcid.org/0000-0002-8963-2893

### Decision letter and Author response

Decision letter https://doi.org/10.7554/eLife.31922.016
Author response https://doi.org/10.7554/eLife.31922.017

## Additional files

### Supplementary files

• Supplementary file 1. List and details of top and bottom magnets.
DOI: https://doi.org/10.7554/eLife.31922.008

• Supplementary file 2. List of TaqMan qRT-PCR assays used in this research.
DOI: https://doi.org/10.7554/eLife.31922.009

• Transparent reporting form
DOI: https://doi.org/10.7554/eLife.31922.010

### Data availability

All data generated or analyzed during this study are included in the manuscript and supporting files.

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
