## [Decision Letter]

[Editors’ note: a previous version of this study was rejected after peer review, but the authors submitted for reconsideration. The first decision letter after peer review is shown below.]

Thank you for submitting your work entitled "Combinatorial Programming of Human Neuronal Progenitors Using Magnetically-Guided Stoichiometric mRNA Delivery" for consideration by *eLife*. Your article has been reviewed by two peer reviewers, and the evaluation has been overseen by a Reviewing Editor and a Senior Editor. The reviewers have opted to remain anonymous.

Our decision has been reached after consultation between the reviewers. Based on these discussions and the individual reviews below, we wish to convey that we find the manuscript of potential interest for *eLife*, but we need to reject the manuscript in its present form because there are too many unresolved technical issues. These issues have been discussed separately in both reviews. Should you be able to address these substantial concern, we would be interested in receiving a revised version of the manuscript.

Reviewer #1:

The manuscript by Azimi et al. describes the development of a new technique for focused transfection of cultured cells that is compatible with combinatorial gene expression studies. They demonstrate the utility of this approach by testing combinations of transcription factors known to be involved in the specification of ventral midbrain dopaminergic neurons. They identified LMX1A, PITX3 and FOXA2 as the most optimal combination of factors and show that PITX3 and FOXA2 factors need to be delivered to dividing progenitors while LMX1A can be delivered to postmitotic neurons. While these three factors were previously shown to promote dopaminergic neuronal identity, this study adds by probing temporal requirements of individual factors.

The major concern with the described method is that it requires the preparation of transfection reagents in large volumes as the entire plate is filled with the solution while only few spots are being transfected. Besides being costly, it might be challenging to perform multiple complete media replacements for truly large scale combinatorial analysis of gene function without disturbing the plated cells. It is not immediately obvious what the principal advantages of the described assay are, compared to preforming the same experiment in a 96 well plate format. The manuscript seems more appropriate for methods journal than *eLife*.

1) The magnetic transfection device and methods need to be described in a greater detail to allow reproduction of the experiments in other labs. For example, the types and provenance of magnets is not obvious. What magnetic fields were tested to achieve optimal transfections and focusing of magnetic particles? Are the top 20mm magnets circular? How does it affect focusing of the particles in the dots aligned with the centers vs spaces between these large magnets?

2) The authors argue that mRNA transfection is superior to plasmid transfection as it allows for better temporal control of gene expression. However no data are provided measuring the kinetics of gene expression – analysis of destabilized GFP or short-lived proteins following magnetofection. Furthermore, the authors should complement their data with a direct comparison of the mRNA vs. plasmid delivery. Demonstrating the use of this technique for simple plasmid transfection is important as long term/stable gene expression might be preferable in many instances. Moreover, the simplicity of plasmid preparation compared to mRNA synthesis would make the technique more generally accessible and interesting to a broader audience.

3) Uniformity of delivered RNA copy-numbers and stoichiometries: Figure 2B shows that the 4 pulses of transfection achieve higher level of GFP and RFP expression. The apparent increase of transfection homogeneity shown in Figure 2C could therefore simply be a consequence of higher reporter expression with more cells reaching saturation. The authors need to compensate the exposure to achieve comparable overall intensity and saturation (i.e. shorter exposure of the 4x experiment) and replot GFP:RFP ratios to more realistically determine whether better homogeneity of expression has been achieved.

4) The authors should comment on the way they switch media between magnetofections – does 4x mean 4x change of media or 4x magnet application with the same media? Do media changes result in neuronal detachment and might that influence final cell counts? Overall it is not clear whether this tool, as described, could be used for "massively combinatorial screening". Experiments performed with few transcription factors do not demonstrate massive parallel screening and therefore references the language should be appropriately toned down.

5) Related to point 2, how long are exogenous FOXA2 and PITX3 proteins detectable in transfected cells and at what point are endogenous factors induced? The authors should epitope tag transfected factors to distinguish them from the induced endogenous factors and probe kinetics of their expression.

6) The text is in places too speculative and hard to follow, there are several typos and panel in Figure 3B does not agree with Figure 3D (see values for PtM vs. PtP, FoM vs. FoP) – very confusing…

Reviewer #2:

This is an interesting technical development for the field. Transcription factors control cell differentiation during embryonic development, thus an appealing tool to guide in vitro cell differentiation for clinical applications. However, these transcription factors work not only in combination but also at different times during development. Thus, to achieve proper and precise in vitro cell differentiation guided by transcription factors it is not only necessary to express combinations of factors but also to do so at different differentiation stages. The method presented by Azimi and co-workers has the potential to accomplish both tasks. Moreover, it can do so in a scalable and multiplexing manner conducive for genetic screens. The value of this paper is the technology and its potential. Thus, I believe that it deserved a better description of the method. The proof of principle for generating TH positive neurons is extremely welcome and elevates the manuscript. However, it will require more experiment to convincingly show that these cells are actual midbrain dopaminergic that are beyond the main innovation presented here. However, a minimal extra quantification will elevate the claims.

1) A better description of the magnet control is necessary for a general audience. Also, a picture or a more desirable video in supplementary data will help to visualize the device and its function. This is a key development and should be understandable by all possible future users.

"'inactivating' unwanted magnets via pushing them significantly down along the z-axis". Pushing the top magnet down does not displace them away from the substrate. Unless this refers to the control of the bottom magnets before the plate is inserted, I am confused by this statement: "The displacement of a magnet as little as 5 mm in z-axis away from the substrate surface is sufficient to eliminate the effect of magnet's magnetic field on the spotting field." Is there a single z-axis manipulator per bottom magnet? Or a single manipulator that moves in the x-y space pushes one magnet at the time? The figure suggests a big plate pushing all the pins.

2) The transfection efficiency and consistency is a key factor in the success of this type of approaches. Thus, an important aspect of this technology to be scalable.

It is clear that a large fraction of the cells express both fluorescent proteins in the improved protocol (Figure 2B). It would be nice to complement Figure 2C with a quantification of the percentage of transfected cells. The authors mention "90%" in the text, thus I assume they have measured. Reporting the actual% over DAPI and its error will provide an estimation of the variability of the method. By the same token, is the PCC calculated from a single transfection or with data from multiple days?

3) I could not find any experiment that validates the expression of other transcription factor other than an increase in FoxA2 over the already high percentage of cells that endogenously express it during differentiation. This manuscript will benefit from a simple validation of at least a couple of them with by antibody staining.

4) This comment is related to the comment above. A potential transformative feature of this new technology is the ability to introduce transcription factors at different states of differentiation, in particular postmitotic stage. Since current methods have limitations, the ability to transfect postmitotic neuros at high efficiency is a highly desirable feature of any protocol. However, I see no evidence that this method actually transfects postmitic neurons. Thus, although the temporal claims in Figure 3 are supported by the experiment, the labeling between mitotic and postmitotic are not. There are very simple experiments to address this point.

5) Dopaminergic neurons are characterized by the expression of several genes. Although required, TH expression by no means is indicative of dopaminergic fate. Thus, additional stainings such as TH in combination with Dbh or Tph2 will demonstrate better that these cells are DA neurons. Also, since the authors have the cDNA, qPCR quantification of other monoaminergic genes will enhance the claims of DA fate over other cell types (i.e. Gad1, Vglut, Sert, Dat, Net, Dbh, Tph).

6) Figure 3B and Figure 3D do not agree. In Figure 3B Fom and Ptm increase TH staining while in Figure 3D Fop and Ptp increase TH. This discrepancy needs clarification since it concerns the conclusion of each factor's activity period to induce TH. I believe this discrepancy also extends to Lmx1b.

7) Figure 2D is beautiful. Please add the concentration for each axis and describe the method. Is this with the single transfection per concentration or it requires interleaved (4x) per concentration? Or alternatively, 4x for GFP is 4x interleaved vs. 1x was a unique event?

[Editors’ note: what now follows is the decision letter after the authors submitted for further consideration.]

Thank you for submitting your article "Combinatorial Programming of Human Neuronal Progenitors Using Magnetically-Guided Stoichiometric mRNA Delivery" for consideration by *eLife*. Your article has been favorably evaluated by a Senior Editor and two reviewers, one of whom, Sacha B Nelson, is a member of our Board of Reviewing Editors.

The reviewers have discussed the reviews with one another and the Reviewing Editor has drafted this decision to help you prepare a revised submission. In considering the nature of this contribution, we suggest this should be viewed as a Tools and Resources paper rather than in the category of a Research Article. That will not affect any aspect of the way this paper is published should the revised version be accepted by the Board.

Summary:

The authors describe a new magnetic spotting technology that permits much more reproducible transfection of cultured cells. By directly transfecting mRNA, they greatly reduce the variability typically associated with plasmid transfection, and by using a device that permits simultaneous magnetotransfection in a multi well format they present a system suitable for high throughput studies. Using this system they demonstrate optimized combinations of transcription factors for generating dopaminergic neurons from stem cells. Although technical in nature, this seems an important development that will help put the field of stem cell biology on a more quantitative footing, a development that is sorely needed.

Essential revisions:

1) Both reviewers felt the comparison with standard transfection methods was unclear. There is no direct comparison to other methods. An image presented in the reply to reviewers makes the essential point, but this point is likely to be lost on most readers of the paper, even including many who routinely use transfection, but do not try to quantify its reproducibility. The correlation coefficients and MOC values obtained seem impressive, but many readers may have no idea how this compares to other methods. The authors should either use some of their own experiments to document the improvement that results from a) RNA over DNA transfection and b) magnetotransfection, or they should use some published data on reproducibility and then analyze their own data in such a manner as to allow comparison. I do not think it necessary to quantitatively assess the two components (RNA and the controllable magnets) separately, although some indication of their relative importance should be provided. Presumably the main reproducibility factor is the RNA and the magnets just allow the neat trick of multiplexing factors and/or dosages.

The fact that the RNA is more reproducible because it is not subject to the multiplicative effect of transcription was not made directly. The focus on numbers of molecules transfected is confusing, since as pointed out by one reviewer, plasmids are not so much larger than the modified RNA molecules.

Along the same lines, it would be helpful to provide benchmarks for readers unfamiliar with current state of the art in differentiating stem cells into dopamine neurons. How do the current results compare? How much of an improvement is this? This is very hard to glean without going through all of the cited papers and other relevant literature.

2) There is not much in the way of consideration of limitations. Most notably, is the method known to be applicable to postmitotic cells? If not, the authors should be clearer on this point. The authors seem to imply that they do not really know if the cells are postmitotic or not when the "postmitotic" transfection is performed. Does the nuclear membrane need to break down for high efficiency? This won't affect experiments on stem cells and cell lines but the methods are potentially also applicable to differentiated cells and whether or not this is the case should be communicated. This point was also raised by a prior reviewer, but was not directly addressed.

The authors do not need to do additional experiments to address this point. They only need to state more clearly what they do and do not know about the transfection. If they have no existing data on truly post mitotic cells, they should take care not to imply that the cells they are transfecting are post mitotic unless they are sure that they are.

[Editors’ note: minor issues and corrections have not been included, so there is not an accompanying Author response.]

Thank you for submitting your article "Combinatorial Programming of Human Neuronal Progenitors Using Magnetically-Guided Stoichiometric mRNA Delivery" for consideration by *eLife*. Your article has been reviewed by a Reviewing Editor in consultation with a Senior Editor.

The many changes made in the manuscript adequately address all of the concerned raised in earlier reviews with one exception. The opening statement "The human nervous system contains hundreds of distinct subtypes of neurons" is in my opinion a vast underestimate and should minimally be qualified with "at least". In addition, the reference cited (#1) does not make any statement about the numbers of cell types in the human or any other nervous system, hence I believe citing it in this context is incorrect.

---

## [Author Response]

[Editors’ note: the author responses to the first round of peer review follow.]

Reviewer #1:The manuscript by Azimi et al. describes the development of a new technique for focused transfection of cultured cells that is compatible with combinatorial gene expression studies. They demonstrate the utility of this approach by testing combinations of transcription factors known to be involved in the specification of ventral midbrain dopaminergic neurons. They identified LMX1A, PITX3 and FOXA2 as the most optimal combination of factors and show that PITX3 and FOXA2 factors need to be delivered to dividing progenitors while LMX1A can be delivered to postmitotic neurons. While these three factors were previously shown to promote dopaminergic neuronal identity, this study adds by probing temporal requirements of individual factors.The major concern with the described method is that it requires the preparation of transfection reagents in large volumes as the entire plate is filled with the solution while only few spots are being transfected.

Our method actually substantially reduces the amount of transfection reagents required per experiment and allows many more experiments to be performed per area. We now explain this more clearly in the manuscript:

"Importantly, with our technology the amount of RNA and transfection reagent required for a given transcription factor is proportional only to the number of spots actually transfected with that factor, rather than to the total number of possible spots in the array (i.e. to the surface area of transfected cells, not to the total surface area of the plate). […] The number of cell medium changes we use also scale linearly with the number of transcription factors delivered, rather than growing exponentially with the number of combinatorial possibilities."

Thus, very small amounts of reagents are spotted specifically onto defined millimeter size spots of cells as opposed to treating entire wells with only a single combination of factors and reagents (for instance, in the experiments shown in our manuscript we use only 0.6 ng RNA for each 1.5 mm diameter spot). Figure 1C and Figure 2D show how the reagents are focally localized onto these very small regions for transfection, which is much smaller than 96- or even 384-well plate formats. In addition, uniform cell culture in 96/384-well plates is often not achievable due to surface effects, a problem that is well known within the high-throughput cell culture screening field.

Besides being costly, it might be challenging to perform multiple complete media replacements for truly large scale combinatorial analysis of gene function without disturbing the plated cells.

We do indeed demonstrate that we can perform multiple rounds of transfections and media exchanges without disturbing the cells or spotting pattern (Figure 2 and Figure 3). We have now made this point clearer in the manuscript: As noted in our response to the previous question, the number of media changes scale linearly with the number of transfection factors while the number of transfected combinations (hence spots) scale exponentially. Thus, for even larger-scale experiments, the number of exchanges would not be much different than what we already do. In addition, industrial systems that change cell-culture mediums from multi-well plates use automated slowly tilting/suction mechanisms for many sequential medium. Although we did not need to implement this in our screen, such systems could be used if cell displacement becomes an issue during manual medium exchange.

It is not immediately obvious what the principal advantages of the described assay are, compared to preforming the same experiment in a 96 well plate format.

As noted above and also made clearer in the revised manuscript, our method’s advantages with respect to 96-well plate formats are tremendous and multifold:

1) Performing experiments on much smaller spots of cells results in a significant savings in reagents. To deliver proportional amount of RNA to each spot comparable to the standard non- spotting transfection, we use an RNA amount linearly proportional to only the number of spots to be transfected and to the surface area of each spot (for instance, 0.6 ng RNA was used for each spot with 1.5 mm diameter). The number of cell medium changes we use also scale linearly with the number of transcription factors delivered, rather than growing exponentially with the number of combinatorial possibilities.

Figure 1c shows how the reagents are focally localized onto a very small footprint for transfection, which is much smaller than 96-well plate formats. In addition, uniform cell culture in 96-well format plates is often not achievable due to surface effects, as known by most people with expertise in high-throughput cell culture screens. Our method thus saves large amounts of transfection reagents by allowing many more experiments to be performed per area.

Here is an estimate of the gain with our technology:

Amount of RNA needed per condition:

96-well plate: 10.9ng (= 0.6ng*(6.4mm/1.5mm)^2^) where 6.4mm is the diameter of a 96- well plate and 1.5mm is the diameter of a spot.

Spotting technology: 0.6ng per spot/condition

Improvement: 18.2 (i.e. ~18 fold)

Amount of transfection reagents (lipid/magnet nano-mixture) needed per combination: Transfection reagents needed scale with the amount of RNA that needs to be transfected. Thus:

Improvement: 18.2 (i.e. ~18 fold)

Amount of cell-culture medium + growth factors needed per combination:

96-well plate: ~6mm x pi*(6.4mm/2)^2^ assuming 200ul per well.

Spotting technology: ~6mm x (3mm)^2^ where 3mm is the spot to spot distance (This estimate includes the unused volume in between spots).

Improvement: 3.6 (i.e. 3.6 fold)

2) The mRNA transfection procedure we developed dramatically reduces fluctuations in copy numbers and expression of delivered factors as we explained in the manuscript in great detail. Author response image 1 clearly shows the variation in gene expression when a 1536-well plate is transfected with DNA (unpublished image provided by Inglese, J. and Jang, S.-W. from NIH Screening Center). Note that each well of a 1536-well plate is even larger than our spots. The left panel shows plating density of a clonal line stably expressing bioluminescent-reporter gene (luciferase) to highlight the well-to-well variation in gene expression on the right panel with transient transfection by luciferase. The reason for this significant variability with DNA transfection is that the number of DNA molecules delivered is typically very low (unlike mRNA transfection). As a result, the Poisson noise in the delivered copy numbers (assuming each transfection process is independent) is huge because of the low mean DNA copy number per cell. This is in contrast to mRNA transfection, where typically hundreds to thousands of copies of the mRNA are delivered during each transfection.

**Author response image 1. respfig1:** Source: Courtesy of Inglese, J. & Jang, S.-W. from NIH Screening Center.

3) The cell cultures in 96-well or smaller formats significantly suffer from edge effects and the culture becomes no longer homogeneous in properties. This challenge is well known within the high-throughput cell culture screening field as shown in Figure 2C from Lundholt et al. (Journal of Biomolecular Screening 8(5); 2003).

The manuscript seems more appropriate for methods journal than eLife.

We believe our manuscript is of great interest not only to the neuroscience and stem cell community, but also to any field of biology involving high-throughput experiments. Many colleagues in the field have expressed great interest in our work. Indeed, NIH Director Francis Collins has highlighted our platform’s blueprint in his presentations among the top examples of high-impact research areas that they fund (This technology is the core of our NIH Director’s Pioneer Award (DP1)).

In addition, we note that *eLife* has a “Tools and Resources” category. According to the *eLife* website: “This category highlights tools or resources that are especially important for their respective fields and have the potential to accelerate discovery. For example, we welcome the submission of significant technological or methodological advances.”

1) The magnetic transfection device and methods need to be described in a greater detail to allow reproduction of the experiments in other labs. For example, the types and provenance of magnets is not obvious. What magnetic fields were tested to achieve optimal transfections and focusing of magnetic particles? Are the top 20mm magnets circular? How does it affect focusing of the particles in the dots aligned with the centers vs spaces between these large magnets?

We have included additional details in the revised manuscript and highlighted the type of magnets used and their shapes:

"We used neodymium (NdFeB) rare-earth magnets because of their high magnetic field strength compared to other types of magnets. […] The magnetic field density on the surface of each magnet is B~300 mT in isolation and B~100 mT when inside the array."

In addition, we have added Supplementary file 1, which contains vendor names and catalog numbers for both top and bottom NdFeB magnets.

2) The authors argue that mRNA transfection is superior to plasmid transfection as it allows for better temporal control of gene expression. However no data are provided measuring the kinetics of gene expression – analysis of destabilized GFP or short-lived proteins following magnetofection.=

With standard DNA delivery methods, genes are forcibly expressed for as long as the DNA template is present in the cell, thus overriding the cell's endogenous mRNA/protein degradation mechanisms. There are chemically controlled gene expression systems that can overcome these concerns, however these are not scalable for high-throughput combinatorial screening. In addition, when using DNA plasmids and/or viral vectors, there is greater uncertainty about the time lag between the transfection and when the desired levels of expression have actually been achieved. With the synthetic mRNAs, both the timing and the relative amounts (i.e. stoichiometry) of each factor are very precise.

Author response image 2 comes from a previous publication by our lab (Figure 1C from Angel and Yanik PLoS ONE 2010) and shows the lifetime of several proteins following transfection of cells with mRNAs that use the same 5’- and 3’ UTRs as those in our current manuscript. These data show the kinetics of protein expression and confirm that tight temporal regulation is possible for most proteins. The exact kinetics of expression will, of course, always vary to some extent between different proteins. We do not believe that quantifying additional specific cases (e.g. destabilized GFP or other proteins) will add significantly to the generality of our platform.

**Author response image 2. respfig2:** Western blots showing expression levels and lifetimes of Oct4, *Sox2*, Nanog, Lin28, and MyoD1 proteins in MRC-5 human fetal lung fibroblasts transfected with protein- encoding RNA, relative to levels in hES (H9) and rhabdomyosarcoma (Rh30) cells. b-actin was used as a loading control. Left panels: The amount of RNA per 50 mL electroporation volume was varied as indicated. Cells were lysed 6 hours after transfection. Right panels: Cells were transfected with 1 mg of RNA, and lysed at the indicated times. (Reproduced from Angel and Yanik, PLoS ONE 2010 under the Creative Commons Attribution License https://doi.org/10.1371/journal.pone.0011756)

To further address the kinetics of transfected mRNAs, in Author response image 3 we provide unpublished RT-PCR data on the lifetime of EYFP mRNA following transfection (again with the same 5’- and 3’- UTRs used in the present manuscript and in Angel and Yanik, 2010). These data show that the mRNAs significantly degrade within 48 hours, a shorter period than the timescale of our cellular programming experiments.

**Author response image 3. respfig3:** 

Furthermore, the authors should complement their data with a direct comparison of the mRNA vs. plasmid delivery. Demonstrating the use of this technique for simple plasmid transfection is important as long term/stable gene expression might be preferable in many instances. Moreover, the simplicity of plasmid preparation compared to mRNA synthesis would make the technique more generally accessible and interesting to a broader audience.

Reliable DNA transfection is not feasible when scaled down to the dimensions we use for spotting due dramatic fluctuations in expression. Each well of the 1536-well plate on the right in Author response image 1 (unpublished data provided with permission of Inglese, J. and Jang, S.-W. from NIH Screening Center) has been transfected with DNA encoding luciferase, clearly showing the variation in gene expression that results under these conditions. The left panel shows a clonal line stably expressing luciferase to better highlight the well-to-well variation in expression on the right panel. Each of our spots is even smaller than the surface area of a single well of a 1536-well plate. The reason for this significant variability with DNA transfection is that the number of DNA molecules delivered is typically very low (unlike mRNA transfection). As a result, the Poisson noise in the delivered copy numbers (assuming each transfection process is independent) is huge because of the low mean DNA copy number per cell. This is in contrast to mRNA transfection, where typically hundreds to thousands of copies of the mRNA are delivered during each transfection.

3) Uniformity of delivered RNA copy-numbers and stoichiometries: Figure 2B shows that the 4 pulses of transfection achieve higher level of GFP and RFP expression. The apparent increase of transfection homogeneity shown in Figure 2C could therefore simply be a consequence of higher reporter expression with more cells reaching saturation. The authors need to compensate the exposure to achieve comparable overall intensity and saturation (i.e. shorter exposure of the 4x experiment) and replot GFP:RFP ratios to more realistically determine whether better homogeneity of expression has been achieved.

This is a good point and we have now explained in the text why this is not the case. Among all conditions in all the images, our camera software automatically finds the brightest spot (across all the well images) and reduces exposure to prevent saturation according to that maximum intensity spot. Thus, the exposure durations in the wells/images used in the generation of the statistics in Figure 2C were not saturated. In addition to clarifying this in the manuscript, we also provided a histogram of pixel intensity distributions of the raw image data in Author response image 4.

**Author response image 4. respfig4:** 

4) The authors should comment on the way they switch media between magnetofections – does 4x mean 4x change of media or 4x magnet application with the same media? Do media changes result in neuronal detachment and might that influence final cell counts? Overall it is not clear whether this tool, as described, could be used for "massively combinatorial screening". Experiments performed with few transcription factors do not demonstrate massive parallel screening and therefore references the language should be appropriately toned down.

The 4x indicates that the medium was changed 4 times. Although we performed many medium exchanges, we do not observe any significant cell detachment in the experiments reported here. We do indeed demonstrate that we can perform multiple rounds of transfections and media exchanges without disturbing the cells or spotting pattern (Figure 2 and Figure 3). The number of media changes scale only linearly with the number of transfection factors but the number of transfected combinations (hence spots) scale exponentially with the number of transfection factors. Thus, for even larger-scale experiments, the number of exchanges would not be much greater than what we already do.

In addition, industrial systems that change cell-culture mediums from multi-well plates use automated slowly tilting/suction mechanisms for many sequential medium exchanges. Although we did not need to implement this in our screen, such systems could be used if cell displacement becomes an issue during manual medium exchange.

We respectfully disagree that our screen was small: we have screened every combination (up to triplets) of 7 transcription factors at different time points. Besides this, we had to repeat/replicate many steps of our experiments during the development of the methodology and also to show the robustness of our results. A much larger screening set-up would be beyond our resources. Our results are sufficient to demonstrate that our platform can be expanded for larger and more elaborate screening procedures.

5) Related to point 2, how long are exogenous FOXA2 and PITX3 proteins detectable in transfected cells and at what point are endogenous factors induced? The authors should epitope tag transfected factors to distinguish them from the induced endogenous factors and probe kinetics of their expression.

Such tracking of endogenous vs. exogenous protein degradation kinetics is very rarely demonstrated in reprogramming studies in the literature (unless it is necessary for the biological question under investigation). Such a study would be beyond the scope of our manuscript.

6) The text is in places too speculative and hard to follow, there are several typos and panel in Figure 3B does not agree with Figure 3D (see values for PtM vs. PtP, FoM vs. FoP) – very confusing.

We have fixed the two typos in the figures and edited the text throughout to improve clarity and readability. If the reviewer could specifically point out any other hard-to-follow or speculative points, we will be happy to further improve them.

Reviewer #2:This is an interesting technical development for the field. Transcription factors control cell differentiation during embryonic development, thus an appealing tool to guide in vitro cell differentiation for clinical applications. However, these transcription factors work not only in combination but also at different times during development. Thus, to achieve proper and precise in vitro cell differentiation guided by transcription factors it is not only necessary to express combinations of factors but also to do so at different differentiation stages. The method presented by Azimi and co-workers has the potential to accomplish both tasks. Moreover, it can do so in a scalable and multiplexing manner conducive for genetic screens. The value of this paper is the technology and its potential. Thus, I believe that it deserved a better description of the method. The proof of principle for generating TH positive neurons is extremely welcome and elevates the manuscript. However, it will require more experiment to convincingly show that these cells are actual midbrain dopaminergic that are beyond the main innovation presented here. However, a minimal extra quantification will elevate the claims.1) A better description of the magnet control is necessary for a general audience. Also, a picture or a more desirable video in supplementary data will help to visualize the device and its function. This is a key development and should be understandable by all possible future users."'inactivating' unwanted magnets via pushing them significantly down along the z-axis". Pushing the top magnet down does not displace them away from the substrate. Unless this refers to the control of the bottom magnets before the plate is inserted, I am confused by this statement: "The displacement of a magnet as little as 5 mm in z-axis away from the substrate surface is sufficient to eliminate the effect of magnet's magnetic field on the spotting field." Is there a single z-axis manipulator per bottom magnet? Or a single manipulator that moves in the x-y space pushes one magnet at the time? The figure suggest a big plate pushing all the pins.

We have rewritten the description of the setup in the manuscript in order to address these concerns. We have also significantly modified Figure 1D to clearly show the sequence of operations. We now use arrows to indicate the direction of motion and the sequence of operation for every moving component in the system.

2) The transfection efficiency and consistency is a key factor in the success of this type of approaches. Thus, an important aspect of this technology to be scalable.It is clear that a large fraction of the cells express both fluorescent proteins in the improved protocol (Figure 2B). It would be nice to complement Figure 2C with a quantification of the percentage of transfected cells. The authors mention "90%" in the text, thus I assume they have measured. Reporting the actual% over DAPI and its error will provide an estimation of the variability of the method. By the same token, is the PCC calculated from a single transfection or with data from multiple days?

We calculated Pearson’s Correlation Coefficient (PCC) and Mander’s Overlap Coefficient (MOC) from different wells with single transfection, and we averaged it over 3 different experiments, which is now indicated in the revised manuscript.

For the 1X transfection protocol 90.4% (236/261) of the detected cells express GFP and 92.3% (241/261) express mCherry. For the 4X interleaved transfection protocol 100% (235/235) of the detected cells express both GFP and mCherry (at varying ratios). The average 90% transfection efficiency was based on determining the number of cells that only express one of the two fluorescent proteins (GFP and mCherry) above a minimum threshold; these cells were considered "untransfected" with respect to the low-expressing protein. Specifically, cells were considered untransfected for a given fluorescent protein if they failed to express it at a level ≧10% of the strongest-expressing cell in the total population analyzed.

3) I could not find any experiment that validates the expression of other transcription factor other than an increase in FoxA2 over the already high percentage of cells that endogenously express it during differentiation. This manuscript will benefit from a simple validation of at least a couple of them with by antibody staining.

We have now clarified this point in this manuscript and in the legend for Figure 4B, which addresses this concern. Although the increase in total FoxA2 is marginal (FoxA2/DAPI in the left panel of Figure 4B), the percentage of FoxA2 cells that also co-express TH (FoxA2/TH in the right panel of Figure 4B) is more than doubled when we deliver the mix of transcription factors FoxA2m, Pitx3m, and Lmx1ap. The quantification based on FoxA2/TH co-expression has been used as the sole criteria in the seminal papers of Lorenz Studer and colleagues.

We have also attempted staining with Girk2, VMAT2, and Lmx1a antibodies in the past, however we found staining by these antibodies to be unreliable over large numbers of samples, even though we tested antibodies from several vendors.

4) This comment is related to the comment above. A potential transformative feature of this new technology is the ability to introduce transcription factors at different states of differentiation, in particular postmitotic stage. Since current methods have limitations, the ability to transfect postmitotic neuros at high efficiency is a highly desirable feature of any protocol. However, I see no evidence that this method actually transfects postmitic neurons. Thus, although the temporal claims in Figure 3 are supported by the experiment, the labeling between mitotic and postmitotic are not. There are very simple experiments to address this point.

"Mitotic" here more specifically refers to the presence of mitogens in the medium and "post- mitotic" specifically refers to the medium condition where mitogens have been withdrawn. To be more specific, we have now clarified what we mean by mitotic and post-mitotic in the manuscript:

"We used two different mediums: 1) an initial expansion medium containing the mitogens bFGF and EGF to sustain the mitotic (proliferative) state and 2) a subsequent differentiation medium lacking the mitogens, causing immediate exit from the cell cycle (post-mitotic state) and inducing neurogenesis."

It is very well known that NPCs require mitogens to proliferate and a standard method to induce differentiation of NPCs is mitogen withdrawal. Although we did not directly measure the mitotic index (e.g. ki-67 staining), we did not observe any additional growth of the cell culture after mitogen withdrawal.

5) Dopaminergic neurons are characterized by the expression of several genes. Although required, TH expression by no means is indicative of dopaminergic fate. Thus, additional stainings such as TH in combination with Dbh or Tph2 will demonstrate better that these cells are DA neurons. Also, since the authors have the cDNA, qPCR quantification of other monoaminergic genes will enhance the claims of DA fate over other cell types (i.e. Gad1, Vglut, Sert, Dat, Net, Dbh, Tph).

Lmx1a and FoxA2 expression are essential to ventral medial DA neuronal progenitors and Nurr1 and Pitx3 are required for the maturation of ventral medial DA neuronal progenitors to mature DA neurons. We chose these in addition to VMAT2 and OTX2 as markers to demonstrate increased dopaminergic fate in vitro by the delivery of RNA factors.

In addition to TH expression, we also observe co-expression of FoxA2 (FoxA2/TH in the right panel of Figure 4B). Although the increase in total FoxA2 is marginal (FoxA2/DAPI in the left panel of Figure 4B), the percentage of FoxA2 cells that also co-express TH (FoxA2/TH in the right panel of Figure 4B) is more than doubled when we delivered the mix of transcription factors FoxA2^m^, Pitx3^m^, and Lmx1a^p^. The quantification based on FoxA2/TH co-expression and the factors we tested above have been used as the sole criteria in the seminal papers of Lorenz Studer and colleagues.

We have not commonly seen the use of co-staining for either Dbh or Tph2 with TH. We are also not aware of midbrain specific cell types that co-express Dbh or Tph2 with TH.

If the purpose were to exclude the possibility of some aberrant cell types that co-express other genes along with TH, this would require a very large-scale profiling which is beyond the scope of our study.

We had also attempted staining with Girk2, VMAT2, and Lmx1a antibodies in the past, however we found staining by these antibodies to be unreliable over large numbers of samples, even though we tested antibodies from several vendors.

6) Figure 3B and Figure 3D do not agree. In Figure 3B Fom and Ptm increase TH staining while in Figure 3D Fop and Ptp increase TH. This discrepancy needs clarification since it concerns the conclusion of each factor's activity period to induce TH. I believe this discrepancy also extends to Lmx1b.

We thank the reviewer for catching this error. We have now corrected it.

7) Figure 2D is beautiful. Please add the concentration for each axis and describe the method. Is this with the single transfection per concentration or it requires interleaved (4x) per concentration? Or alternatively, 4x for GFP is 4x interleaved vs. 1x was a unique event?

Figure 2D involves 24 medium exchanges [interleaved (4x) per concentration] to generate all the possible combinations. We have now clarified this in the manuscript. We now also indicate concentration on the figure axis.

[Editors' note: the author responses to the re-review follow.]

Essential revisions:1) Both reviewers felt the comparison with standard transfection methods was unclear. There is no direct comparison to other methods. An image presented in the reply to reviewers makes the essential point, but this point is likely to be lost on most readers of the paper, even including many who routinely use transfection, but do not try to quantify its reproducibility. The correlation coefficients and MOC values obtained seem impressive, but many readers may have no idea how this compares to other methods. The authors should either use some of their own experiments to document the improvement that results from a) RNA over DNA transfection and b) magnetotransfection, or they should use some published data on reproducibility and then analyze their own data in such a manner as to allow comparison. I do not think it necessary to quantitatively assess the two components (RNA and the controllable magnets) separately, although some indication of their relative importance should be provided. Presumably the main reproducibility factor is the RNA and the magnets just allow the neat trick of multiplexing factors and/or dosages.

We now include references that directly compare RNA- vs. DNA-based transfection approaches head-to-head in various difficult to transfect cell lines with detailed efficiency assays. These references support our claim regarding the superior efficiency and reliability of RNA over DNA. As the reviewers and editor correctly hypothesize, the magnetic transfection reagent (CombiMag) functions to achieve precise localization/multiplexing of RNA molecules, while the transfection process itself relies on a commercial transfection reagent rather than the magnets. We have now re-written the paragraph discussing the benefits of RNA to emphasize well-known mechanistic differences impacting the expression of exogenous RNAs vs. DNAs:

"The fact that cell fate decisions are highly dependent on the precise expression levels of a limited set of genes demands that each reprogramming factor in a given cocktail needs to be delivered with minimal cell-to-cell variation to achieve maximal efficiency. […] This is supported by the low variability of RNA transfection results with respect to DNA transfection (Hansson et al., 2015; Landi, Babiuk and van Drunen Littel-van den Hurk, 2007)."

However, when using special cell lines that are easily transfected and/or transfection protocols that have been optimized for delivery of DNA, it may indeed be feasible to apply our platform to combinatorial screens that use plasmids or other DNA-based vectors. The primary emphasis of our manuscript is on validating our high-throughput platform for magnetically-guided combinatorial transcription factor screens, rather than exploring all possible cell types, nucleic acid vectors, and/or commercial transfection reagents that our platform may be compatible with. It has been established in the literature that CombiMag can be used to deliver DNA (Varro, Kenny et al., 2007), siRNA (Lee, Shim et al., 2011) miRNA (Zhang, Tang et al., 2016), retroviruses/lentiviruses (Fukushima, Tezuka et al., 2007), and even proteins (Watanabe, Tatebe et al., 2012) to a variety of cell types. In addition, it has been demonstrated to function with a variety of commercially available lipid-based transfection reagents.

The fact that the RNA is more reproducible because it is not subject to the multiplicative effect of transcription was not made directly. The focus on numbers of molecules transfected is confusing, since as pointed out by one reviewer, plasmids are not so much larger than the modified RNA molecules.

As noted above, although plasmids are not substantially larger than RNA molecules, the fact that they need to cross both the plasma membrane *and* the nuclear envelope can have a profound impact on the number of DNA molecules that can be functionally expressed. This is particularly true for difficult to transfect cell types [as noted in the newly incorporated Hansson et al. (2015) and Landi et al. (2007) references]. It is also a consideration that is generally acknowledged in the technical literature and online documentation accompanying commercial transfection reagents (for example: https://www.biocompare. com/Editorial-Articles/171593-What-to-Transfect-DNA-vs-RNA-vs-Protein/). Therefore, the variation observed between RNA and DNA transfections is likely not due exclusively to the fact that DNA is subject to transcriptional multiplication, but also due to the relatively smaller number of DNA molecules that are expressed.

Along the same lines, it would be helpful to provide benchmarks for readers unfamiliar with current state of the art in differentiating stem cells into dopamine neurons. How do the current results compare? How much of an improvement is this? This is very hard to glean without going through all of the cited papers and other relevant literature.

As requested, we have now included an overview of a range of differentiation efficiencies that have been reported in the literature as well as a discussion of why direct head-to-head comparisons between different publications/protocols is often challenging and how our platform can help address this challenge in the future:

"During the past decade, there have been significant efforts to generate dopaminergic neurons from human pluripotent stem cells (hPSCs) (Kriks et al., 2011; Theka et al., 2013; Friling et al., 2009; Lee et al., 2010), typically by supplementing the cell medium with various growth factors. In addition to growth factor-based differentiation protocols, exogenous overexpression of transcription factors (such as the single transcription factor *LMX1A* (Friling et al., 2009) or the combined overexpression of *FOXA2* and *NURR1* (Lee et al., 2010)) using DNA vectors can partially enhance the number of tyrosine hydroxylase (TH) positive dopaminergic neurons during differentiation of pluripotent stem cells under various culture conditions. […] Our high-throughput platform, where many factors and conditions can be simultaneously tested side-by-side and compared under the same laboratory conditions, can in the future allow for more accurate and quantitative comparisons."

2) There is not much in the way of consideration of limitations. Most notably, is the method known to be applicable to postmitotic cells? If not, the authors should be clearer on this point. The authors seem to imply that they do not really know if the cells are postmitotic or not when the "postmitotic" transfection is performed. Does the nuclear membrane need to break down for high efficiency? This won't affect experiments on stem cells and cell lines but the methods are potentially also applicable to differentiated cells and whether or not this is the case should be communicated. This point was also raised by a prior reviewer, but was not directly addressed.The authors do not need to do additional experiments to address this point. They only need to state more clearly what they do and do not know about the transfection. If they have no existing data on truly post mitotic cells, they should take care not to imply that the cells they are transfecting are post mitotic unless they are sure that they are.

Consistent with the published literature, we observe very little proliferation once mitogen growth factors have been removed from the medium. However, in order to avoid these concerns, we have replaced all "mitotic/post-mitotic" terminology in the revised manuscript and now simply refer to whether transfections are being done on proliferative NPCs (i.e. cultured in the presence of growth factors) or after induction of neural differentiation (i.e. after grown factor removal). In the revised versions of Figure 3 and Figure 4 these conditions are indicated with an 'N' or an 'I' superscript, respectively.

Corresponding changes have been made to terminology throughout the manuscript where required. For example, from the Introduction:

"Using our magnetically-guided spotting platform and interleaved transfection protocol, we evaluated the temporal contributions of transcription factor cocktails by treating human NPCs with them during the proliferative stage and/or during the induction of neurogenesis (i.e. after mitogen withdrawal) to generate human dopaminergic neurons with high purity."

And from the Results:

"Transcription factors were delivered individually either during the proliferative NPC stage (Day -2, before mitogen removal) or the induction stage (Day 0, upon mitogen removal) as indicated by the superscripts N or I on each factor, respectively."